# Effect of Enriched Substrate on the Growth of the Sea Cucumber *Holothuria arguinensis* Koehler and Vaney, 1906 Juveniles

Tiago Rodrigues [1,*], Francisco Azevedo e Silva [1,2], João Sousa [1], Pedro M. Félix [2,*] and Ana Pombo [1]

1    MARE—Marine and Environmental Sciences Centre/ARNET–Aquatic Research Network, ESTM, Polytechnic Institute of Leiria, 2520-641 Peniche, Portugal
2    MARE—Marine and Environmental Sciences Centre/ARNET–Aquatic Research Network, Faculdade de Ciências, Universidade de Lisboa, 1740-016 Lisboa, Portugal
*    Correspondence: rodrigues.tiago@gmail.com (T.R.); pmfelix@ciencias.ulisboa.pt (P.M.F.)

**Abstract:** The sea cucumber *Holothuria arguinensis* (Koehler and Vaney, 1906) presents an increasing commercial value in Asian markets and its exportation depends exclusively on wild stock harvesting. Production through aquaculture has been acknowledged as an alternative to supply demand and reduce pressure on natural stocks, but there are still bottlenecks to overcome, such as low growth rate and survival. This study focuses on the evaluation of the growth of juveniles of *H. arguinensis* through substrate enrichment—direct mixing of *Sargassum vulgare* with sediment—in recirculating aquaculture systems (RAS), for 4 months, with a baseline diet of *Saccorhiza polyschides*. Compared to the control (no enrichment), sea cucumbers fed with enriched sediment showed the highest specific growth rate (SGR), the highest growth rate (GR) and the lowest size heterogeneity. The results obtained in this study showed a favorable growth of sea cucumbers when in the presence of a substrate enriched with *Sargassum vulgare*, representing an important step towards the viability of large-scale sea cucumber rearing in Europe.

**Keywords:** holothuroidea; echinoderm; aquaculture; *Sargassum*; diet; growth optimization; captivity





## 1. Introduction

In a context of increasing demand for sea cucumbers and decreasing ability of fisheries to supply the Asiatic markets, the aquaculture production of these species offers a solution to minimize the depletion of natural stocks, while offering alternatives to meet demand. The growing aquaculture of sea cucumbers *Apostichopus japonicus* (Selenka, 1867) and *Holothuria scabra* (Jaeger, 1833) has become a profitable industry while helping the conservation of species [1]. Over the past decades, the culture of *A. japonicus* has become one of the most important aquaculture industries in China, and using the same rearing and cultivation techniques, *H. scabra* has also started to be produced on a commercial scale. However, when it comes to sea cucumber production, Europe is still behind, compared to Asia. Nevertheless, over the years there has been an increasing number of papers published in this area [2–6], where some European sea cucumber species have been demonstrated to be potential aquaculture candidates [2,5–10]. Several of these temperate species are new targets for exploitation and have increasing market values [11]. The sea cucumber *Holothuria arguinensis* is among those candidates with high potential for aquaculture production [8].

As a new target species, *H. arguinensis* has been exploited in the Atlantic and Mediterranean for exportation to Asian markets. The catches driven by its high commercial value are growing [9,12,13]. The development of *H. arguinensis* aquaculture is crucial to face the overexploitation and high fishing pressure on this species, but the shortage of studies that allow an optimal production in aquaculture of this species is hindering the progression. From broodstock maintenance to reproduction, larval development, and juvenile rearing,

the studies on *H. arguinensis* have also been increasing. The production of hatchery-reared juveniles has been achieved and the testing of zootechnical conditions aim for an optimized growth and survival [6]. However, there is still a lack of studies pertaining to juvenile rearing. Furthermore, there has not been any difference in the food application method, where the main focus of most of the studies is the different algae and different types of diets to supply in order to improve the growth of sea cucumbers. From an aquaculture perspective, the feeding process for *A. japonicus* and *H. scabra* has been achieved by simply providing fine algae powder to the production tanks, letting it sink, settling on the sediment. However, no studies have been carried out with a specific focus on using an enriched substrate as way to provide food to any Atlantic sea cucumber species, including *H. arguinensis*. Sea cucumbers, as detritivorous organisms, derive nutritional benefits from sediment complexity. In captivity, this complexity may be achieved through a direct enrichment of the sediment that may increase the homogeneity of food availability across the substrate, but also enable natural biological and chemical processes in the substrate, promoted by bacteria or fungi [14]. With this method, the loss of food in the water column is reduced, and the accessibility to food is achieved in a natural way while the sea cucumbers ingest the substrate. Objectively, this study aimed to evaluate the effects that a substrate mixed with algae (*Sargassum vulgare*) may have on the growth of *H. arguinensis* juveniles. For that, the difference between an enriched substrate and a non-enriched substrate was evaluated for a period of four months, in an experimental design comprising individuals from different size classes. During this period, all individuals were systematically assessed using growth metrics, as a proxy of condition.

## 2. Materials and Methods

### 2.1. Experimental Design

The experiment was conducted in the aquaculture Lab of Mare–Polytechnic of Leiria using 3 recirculating aquaculture systems (RAS), in 3 tanks (A, B, C) with 552 L (2.3 m × 0.6 m × 0.4 m), each divided into 3 sections, using PVC planks ($n = 9$): A1, A2, A3; B1, B2, B3; C1, C2, C3, with 184 L each. Holes were made in the planks allowing the water to flow between divisions of each RAS, covered with a net with 1 mm mesh-size to prevent the escape of any individuals between divisions. To guarantee that the water reached the sump equally from each division, a perforated 1.70 m tube crossed all tanks, ensuring homogeneity of water quality and filtration. Each system was connected to its own sump that included mechanical filtration (wool and sponge filters), biological filtration (plastic bio-balls), a Bubble Magus C3.5 Needle Wheel Protein Skimmer (Jiyang Aquarium Equipment Co., Ltd., Jiangmen, China), and a Hailea HX-6530 water pump (Guangdong Hailea Group Co., Ltd., Guangzhou, China).

The hatchery-reared juveniles used in the trial were previously grown in an RAS system with 0.4 m$^3$ tanks with sediment, equally fed with *Ulva lactuca*. The 4-month experiment was performed from January 2022 to May 2022. After all the procedures to prepare the experiment systems, a fasting period preceded the trial. For that, a total of 108 sea cucumbers were translocated to the experiment systems (A, B and C) without any substrate, at 19 ± 1 °C, where they spent 2 days removing all the gut contents in the digestive system. After these 2 days, all individuals were measured and weighed and assigned to 3 different triplicated weight classes: Class 1 (>8 g, $n = 33$); Class 2 (4–8 g, $n = 36$); Class 3 (<4 g; $n = 39$), ensuring no significant differences between replicates. Replicates were assigned each to a different RAS, making sure each system contained all three weight classes (Figure 1): 11 individuals of Class 1 in each division A1, B1, C1; 12 individuals of Class 2 in each division A2, B2, C2; and lastly 13 individuals of Class 3 in each division A3, B3, C3. The trial was conducted at low density (<1 kg·m$^{-2}$), as high densities increase stress in juvenile sea cucumbers, affecting their metabolism and growth [15]. Individuals distributed between the experimental tanks of the trial presented a non-normal distribution, thus according to a Kruskal–Wallis test, no statistically significant differences in the same size classes ($p = 0.988$) were found, which allowed the trial to begin, and for viable comparisons between groups to be performed at the end of the trial.

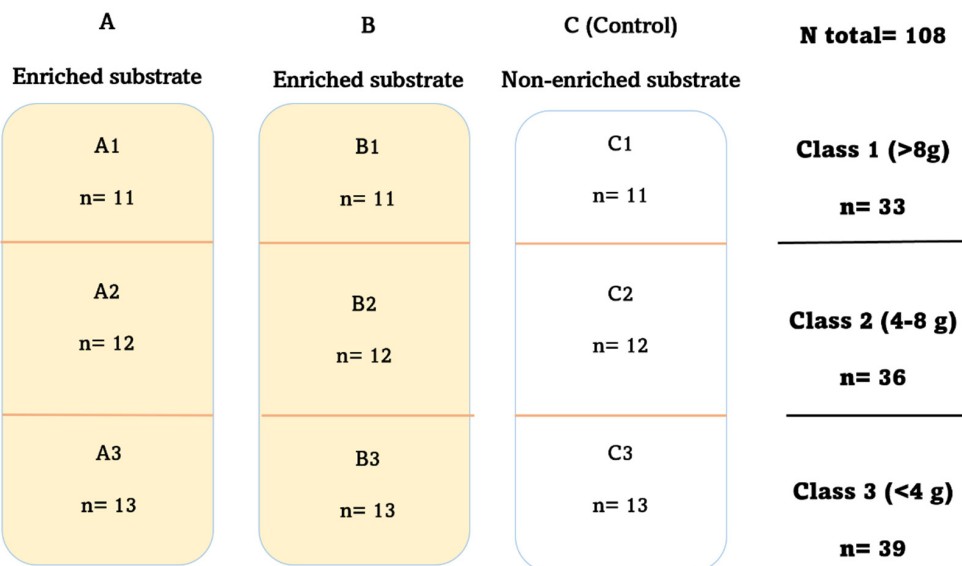

**Figure 1.** Three different RAS systems (A, B, C) with three divisions in each tank (A1, A2, A3, B1, B2, B3, C1, C2, C3). Each division from each tank housed one of the three weight classes of *Holothuria arguinensis* juveniles. System A and B received the enriched substrate mix (sand + macroalgae) while system C received only sand. A baseline diet of *Saccorhiza polyschides* was provided to all systems.

To test the influence of the enriched sediment on juvenile development against a control group, without enrichment, systems A and B (6 divisions) were prepared with the enriched sediment (*S. vulgare*) and system C (3 divisions) without enriched sediment. The enriched sediment was formulated by mixing 8% of blended algae with 92% of sand (*w/w*), defined to represent a percentage of organic content similar to the maximum value in the natural environment of this population [16], and placing it in a total of 6 out of 9 recipients where 3 had no presence of algae. The macroalgae were previously ground into fine fragments, allowing a better mix with the sand. Both recipients, with and without the sand–algae mix, were filled with water, which would soak the algae and sand, allowing an easier settlement in the bottom of the tank by preventing the algae from floating. As a baseline, every group was also fed with *Saccorhiza polyschides* three times per week during the entire experiment. The daily feeding dose was calculated based on the body weight of sea cucumbers in each division separately, rating 3% of body weight [17]. This value was readjusted according to the weight gain of the animals every 14 days during the 4 months of the experiment.

The entire experiment was conducted in a controlled environment, with a photoperiod 12 hL:12 hD. The seawater used in the experiment was previously filtered using mechanical and biological filtration. The measurements of temperature ($^\circ$C), dissolved oxygen (%), pH, $NH_3$ (mg·$L^{-1}$), $NH_4^+$ (mg·$L^{-1}$), and $NO_2^-$ (mg·$L^{-1}$) were taken daily during the trial. Water exchanges were carried out whenever necessary to maintain the water quality. The temperature of the water was 20 $^\circ$C $\pm$ 1. Salinity was maintained in a range of 33.3 $\pm$ 0.1 and dissolved oxygen around 81.6% $\pm$ 0.3, while pH was kept at 8.06 $\pm$ 0.02. Ammonia and nitrites ($NH_3$ and $NO_2^-$ in mg·$L^{-1}$) were measured with API color testing kits (Mars Fishcare) and maintained at 0.03 mg·$L^{-1}$ $\pm$ 0.02 and at 0.00 mg·$L^{-1}$ $\pm$ 0.03, respectively. Every 14 days, matching with the sampling for weighing (fresh weight, FW) and measuring (fresh length, FL) of the individuals, the tanks were emptied. The sediment was totally removed from the tanks, and 90% of the water, with both parts being renewed for a new cycle, guaranteeing similar conditions throughout the experimental period.

Every 14 days (T0 to T9), all sea cucumbers were measured underwater, using a flexible measuring tape, for total length ($\pm$0.1 cm), without any manipulation, avoiding muscular contractions and any bias in this metric [18]. Using a digital scale (ADAM PGL Precision Balance NDPGL4001, EUA) with a precision of 0.01 g, the sea cucumbers were also weighed, and the data registered. The sea cucumbers were placed into the different

recipients (one for each of the 9 groups) filled with water from the system, with the presence of constant aeration. Without the sea cucumbers in the tanks, and after all the cleaning and renovation process, a new mix of sand and *S. vulgare* was placed in the tanks of systems A and B, and new sand in C. At each renewed cycle, a close and continuous follow-up of the water parameters (temperature, salinity, pH, ammonia, nitrites and nitrates) assured stabilizations and no differences between systems. Afterward, the sea cucumbers were placed back in the compartments of the tanks in the correct class divisions.

### 2.2. Data Analysis

Growth performance and survival rate were evaluated by determination of survival rate (SR), specific growth rate (SGR), body length rate (GR). These calculations were supported using the following formulae:

$$\text{SR } (\%) = \frac{final\ number\ of\ animals}{initial\ number\ of\ animails} \times 100$$

$$\text{SGR } (\%/\text{day}) = \frac{\ln(Wf) - \ln(Wi)}{T} \times 100$$

$$\text{GR } (\%\text{cm}/\text{day}) = \frac{Lf - Li}{Li \times T} \times 100$$

where *Wi* is the initial weight (g) and *Wf* is the final weight (g). *Li* is the initial length (cm), and *Lf* is the final length (cm). *T* is the experiment time in days.

The obtained data were used to perform a Kruskal–Wallis one-way analysis of variance, due to the non-normality of the data, in order to accurately determine any differences in the individuals between, both experiment groups (A and B) and, between both the experiment groups and the control (C), before and after the trial. Whenever any statistically significant difference occurred ($p < 0.05$), it was immediately followed by a pairwise multiple comparison procedure (Mann–Whitney) method for non-parametric tests. All statistical analyses were performed with the software IBM SPSS Statistics 28.0.0.1. All data were presented, whenever possible, as mean $\pm$ standard deviation (SD). Relative standard deviation (RSD) was also performed using the weight of the sea cucumbers in the different classes, to determine the evolution of the sample unit's heterogeneity.

## 3. Results

### 3.1. Survival Rate

During the daily routine or sampling moments, any casualties that occurred were registered. At the end of the experiment, the survival rate ranged from 91% to 97%, between systems, showing an overall 93% of survival, as 101 out of the initial 108 individuals reached the end of the experiment successfully. Although no significant differences were found between systems ($p > 0.05$), mortality was mostly registered in system C (5 in total), the control, followed by system A (2 in total). System B showed no casualties in any weight class (Table 1).

**Table 1.** Mortality rate (%) for the different weight classes (1, 2 and 3) and system (A, B—treatment, and C—control) for the feeding trial with enriched sediment with *Holothuria arguinensis*.

|  | Class 1 (>8 g) | Class 2 (4–8 g) | Class 3 (<4 g) | Total % |
|---|---|---|---|---|
| System A | 3.03 | 2.70 | 0 | 5.73 |
| System B | 0 | 0 | 0 | 0 |
| System C | 0 | 5.55 | 7.69 | 13.24 |

### 3.2. Weight Analysis

At the end of the trial there was an increase in fresh weight (FW) in most of the treatments. In terms of classes, the highest mean FW was always found in system B (27.77 $\pm$1.76 g for Class 1; 16.96 $\pm$1.33 g for Class 2; 8.76 $\pm$1.22 g for Class 3). From T0 to T1 there was always a

weight gain (Figure 2), but from T1 to T2, systems A and C, in particular, showed a decrease for Classes 2 and 3. In general, all classes from system B showed a continuous growth until T5, after which it started to stabilize. In system C, there was a decreasing trend in weight after the first month. At the end of the experiment (T9) there was a statistical difference ($p = 0.015$) between the weight of the same class in the different tanks. The pairwise tests for Class 1 showed a significant difference between each treatment group (A1, B1) and the control group (C1) (M–W: $p < 0.001$), and no difference ($p = 0.418$) between the two experimental groups (A1–B1). Classes 2 and 3 showed significant differences between tanks at T9 for all systems (M–W Class 2: A2–C2, $p < 0.001$; B2–C2, $p < 0.001$; A2–B2, $p < 0.001$; and M–W Class 3: A3–C3, $p = 0.008$; B3–C3, $p < 0.001$; and A3–B3, $p = 0.003$).

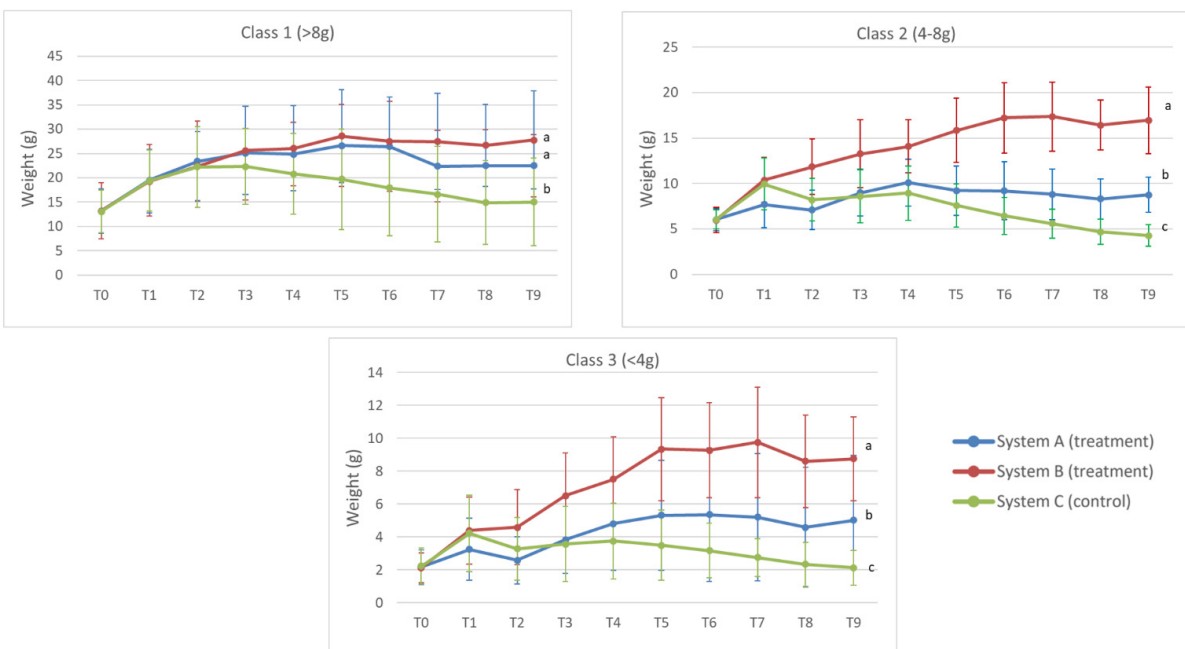

**Figure 2.** Fresh weight (FW) variation for the different weight classes (1, 2 and 3) of *Holothuria arguinensis* in the different systems (A and B for treatment and C for control) with fortnightly sampling times (T0, T1, T2, T3, T4, T5, T6, T7, T8, T9). Different lowercase letters indicate significant difference at $p < 0.05$.

In general, the sea cucumbers of system B showed a higher growth in each different class, as well as a higher SGR for all size classes (Figure 3), whilst individuals from system C presented the lowest SGR, displaying negative values for the smaller weight classes. The sea cucumbers of system B showed the highest SGR (1.13%) in Class 3, while the lowest (−0.27%) was shown in Class 2 of system C.

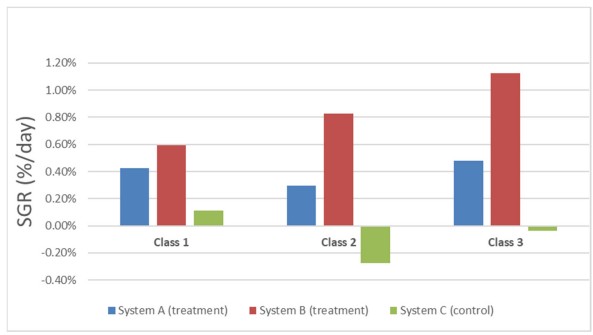

**Figure 3.** Specific growth rate (%/day) for the different weight classes (Class 1 (>8 g), Class 2 (4–8 g) and Class 3 (<4 g)) of *Holothuria arguinensis* in the different systems (A and B for treatment and C for control).

### 3.3. Length Analysis

At the end of the trial, similarly to the final FW, the higher results for fresh length (FL) in each class were obtained in system B ($10.58 \pm 1.76$ cm for Class 1; $9.23 \pm 1.33$ cm for Class 2; $8.76 \pm 1.22$ cm for Class 3), represented in Figure 4. The statistical evaluation at T9 indicated significant differences in the final fresh length between different classes in the different systems ($p = 0.003$). In Class 1, the Mann–Whitney test showed a statistical difference between systems A–C ($p = 0.016$) and B–C ($p = 0.001$) while between the sea cucumbers of systems A–B, no differences were found ($p = 0.099$). Classes 2 and 3 showed significant differences between systems at T9 for all systems (M–W Class 2: A2–C2, $p < 0.001$; B2–C2, $p < 0.001$; A2–B2, $p < 0.001$; and M–W Class 3: A3–C3, $p < 0.001$; B3–C3, $p = 0.003$; and A3–B3, $p < 0.001$).

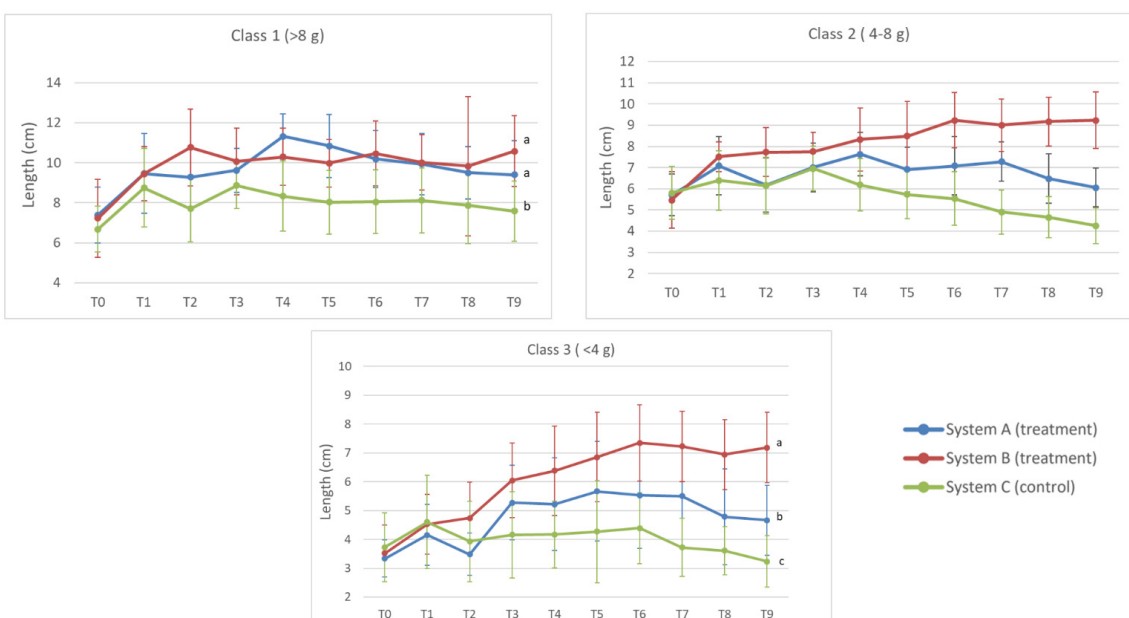

**Figure 4.** Fresh length (FL) variation for the different weight classes (1, 2 and 3) of *Holothuria arguinensis* in the different systems (A and B for treatment and C for control) with fortnightly sampling times (T0, T1, T2, T3, T4, T5, T6, T7, T8, T9). Different lowercase letters indicate significant difference at $p < 0.05$.

Regarding the body length rate (%cm/day), sea cucumbers in system B had the highest Gr in every class (Figure 5). In systems A and B, with the enriched sediment, the higher rate was noticeable, especially in Class 3, with 0.399 and 1.180% cm/day, respectively. While in the control tank the results were low and even negative, showing a minimum value of −0.212% cm/day in Class 2.

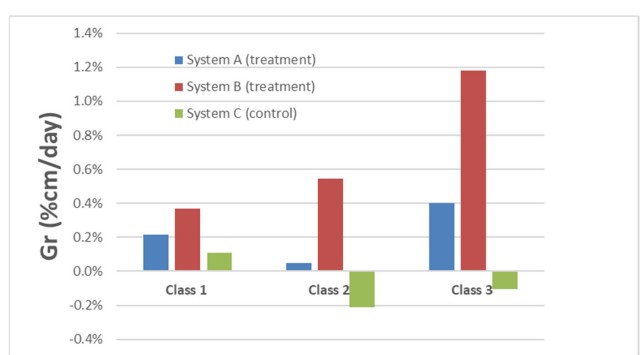

**Figure 5.** Growth rate (%/day) for the different weight classes (Class 1 (>8 g, Class 2 (4–8 g) and Class 3 (<4 g)) of *Holothuria arguinensis* in the different systems (A and B for treatment and C for control).

### 3.4. RSD Analyses

Relative standard deviation (RSD), with the measured weight, was calculated to represent the evolution of size heterogeneity. It revealed the overall best results in system B, for all classes, but particularly in Class 3, where over time the individual sizes became more homogeneous (Figure 6). An increase in heterogeneity was found in the control group for the larger individuals (Class 1) and in system A for the smaller size class.

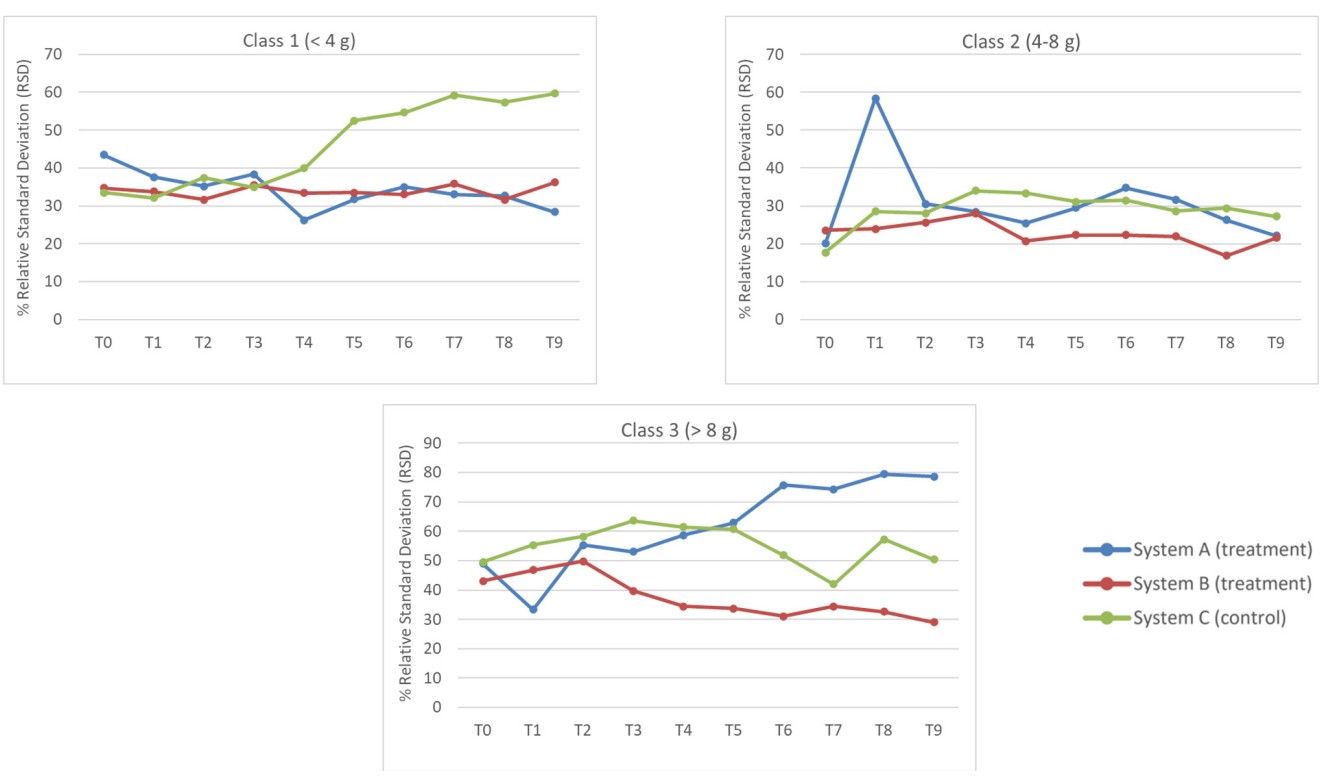

**Figure 6.** Relative standard deviation of *Holothuria arguinensis*, of the mean weight of each size-class (1, 2 and 3) in the different systems (A, B and C) and sampling times (T0 to T9).

## 4. Discussion

Growth and feeding methods of sea cucumbers are optimized for cultivated species but remain understudied for *Holothuria arguinensis*. Different settings have been tested for the growth of sea cucumbers in recent years, from different temperatures [6], different salinities [19] to different substrate sizes [20] and different diets [5], but never using the enriched substrate as a method to provide the food to any Atlantic sea cucumber species, including *H. arguinensis*.

Regarding the present study, the results obtained showed that an enriched substrate influences the growth of *H. arguinensis* in a positive way. No significant differences were found in the mortality between all systems, showing that there were adequate zootechnical conditions for the sea cucumbers physiology. Despite this, the mortality registered in the control system was not expected, even though the feeding method, based only on the addition of macroalgae, could be the justification for the casualties, due to malnutrition and consequent physiological impairment. As described in the Sinsona and Juinio-Meñez [21] study with *H. scabra*, the largest juveniles in the presence of sediment enriched with *Sargassum* sp. showed a significantly higher survival rate when compared to the ones without any enrichment and, also, the smaller individuals with no enriched sediment showed a lower survival rate. In the present study, the highest survival rate was also obtained in the biggest size class with the enriched sediment, and the most mortalities, with a lower survival rate in the smallest size class, was in the control system (without enrichment). However, the 60-day experiment with sediment enriched with *Sargassum* sp.,

conducted by Sinsona and Juinio-Meñez [21], showed a lower survival rate than the present study, at 63.3%. A high survival rate when using *Sargassum* sp. as a diet was also demonstrated in a study performed by Magcanta et al. [22], where the sea cucumbers, *H. scabra,* fed with *Sargassum* showed a 92.67% survival rate within the ideal range of salinity (32–35), showing similar results to those obtained in this study. One justification could be their low mobility. As Olaya-Restrepo [23] showed, sea cucumbers move in order to find a suitable place to feed, which, together with the smaller size of the sea cucumbers found in Classes 2 and 3 of system C, suggests that the process of searching for food may require more energy than that obtained during the feeding process. For this reason, an enriched sediment would provide higher homogeneity in the food placement, allowing the sea cucumbers to find food more easily without unnecessary energy expenditure, unlike individuals in a non-enriched sediment.

The differences in weight found between the control and treatment are clear. As described in most studies involving these organisms with different diets [24–27], a higher growth is commonly observed in the first sampling moments, followed by a stabilization period. For systems A and B, the highest value before the stabilization of growth was noticed at T5 (70 days into the experiment), showing an improvement in the growth rate of the *H. arguinensis*, whereas in the abovementioned studies the stabilization would happen after 30 days of experiment. For the control system, however, the weight values started to lower at T2 (28 days into the experiment), similarly to the previous studies. The higher values of final weight obtained for systems A and B compared to C can, thus, be justified by the higher food availability with the enriched sediment.

In Ria Formosa, higher growth rates of *H. arguinensis* were observed, having as possible justification the high productivity of the ecosystem due to the rich phytobenthos of macrophytic and microphytic organisms (seagrass and diatoms) [23,28]. This could be a possible explanation for the difference found in the results between the sea cucumbers present in the control system (C) and the test systems (A and B). Thus, an enriched sediment may favour the nutritional process, in quantity and quality, as opposed to waiting for supplied food to deposit in the bottom. So, an enriched sediment will not only provide an easy access to food, but also an easy way to fixate benthic species in the sediment, species that provide a complex set of nutritional elements for the diet of deposit-feeders, such as sea cucumbers [29,30]. Sinsona and Juinio-Meñez [21], when testing an enriched sediment for *H. scabra*, obtained a higher weight of 38.69 ± 8.0 g in sea cucumbers placed with the presence of an enriched sediment, showing similar results to those obtained in the present study, where the highest value of weight was also obtained in sea cucumbers with the presence of enriched sediment. The difference between the treatment systems and the control, depicting a higher SGR for the sea cucumbers reared with the enriched sediment and lower for the control, was also found in the Sinsona and Juinio-Meñez [21] study, where the sea cucumbers fed with sediment enriched with *Sargassum* sp. showed the best values of SGR after 30 days of the experiment (4.63%/day). The SGR difference between treatment systems (A and B) does not offer a direct explanation based on food supply or zootechnical parameters that were kept at the same levels. However, system A was the system that needed the most maintenance, since the pvc planks that made the separations became detached from the tank walls in the middle of the experiment. To prevent any escapes from the sea cucumbers to the other divisions in the tank it had to be emptied and the pvc planks had to be glued back to the tank walls on the same day. This manipulation could have induced additional stress on the sea cucumbers, leading to lower growth, justifying the difference between systems A and B.

The relative standard deviation described in this study showed that sea cucumbers maintained in system B had the best results with minor changes between values, showing a more homogeneous growth, and the lowest values obtained during the study period. For individuals maintained in system C, high differences were noticed, especially in Class 1. This shows that, on one hand, low food availability may aggravate size heterogeneity, but on the other hand, an enriched sediment may invert the trend. High heterogeneity

on the growth of sea cucumbers when in captivity is a difficult obstacle to overcome, as demonstrated in Laguerre et al. [31], and can be a production bottleneck in aquaculture, because cohorts tend to attain commercial sizes at different times.

## 5. Conclusions

This study demonstrated that sediment enriched with *S. vulgare* improves the growth of *H. arguinensis*. Combined with the optimal temperature, the enriched sediment can provide an advantage in the production of *H. arguinensis*. This first report of this feeding method for this sea cucumber species is a step towards the future of sea cucumber aquaculture in Europe, and especially in the domestication of *H. arguinensis*. Knowing that the growth is improved by an enriched substrate, future studies should take into consideration the nutritional quality of the enriched sediment, to understand if the nutritional aspect can be a key point, or if the quantity of food and ease of access are the primary reasons for the higher growth rates.

**Author Contributions:** Conceptualization, resources and supervision, P.M.F., A.P. and F.A.e.S.; Lab settings, preparation and testing, T.R., F.A.e.S. and J.S.; Lab trials, T.R., F.A.e.S. and J.S.; Data analysis, T.R.; manuscript writing, T.R.; manuscript review, P.M.F., A.P., F.A.e.S. and J.S. All authors have read and agreed to the published version of the manuscript.

**Funding:** The authors acknowledge the Fundação para a Ciência e Tecnologia (FCT), through the strategic project LA/P/0069/2020 granted to the Associate Laboratory ARNET and UIDB/04292/2020 and UIDP/04292/2020 awarded to MARE-Marine and Environmental Sciences Centre. FAS is supported by a PhD fellowship granted by the FCT (reference SFRH/BD/09563/2020) and AP under the Scientific Employment Stimulus (CEECINST/00051/2018) granted by the FCT.

**Data Availability Statement:** The datasets generated during and/or analysed for the current study may be available from the corresponding author on reasonable request.

**Acknowledgments:** The authors would like to thank the Instituto para a Conservação da Natureza (ICNF) for their collaboration in the field, and also to Pedro Moreira for the help provided in the sample moments.

**Conflicts of Interest:** The authors declare no conflict of interest.

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
