# Peer review of "Effect of Enriched Substrate on the Growth of the Sea Cucumber Holothuria arguinensis Koehler and Vaney, 1906 Juveniles"

_diversity, doi:10.3390/d15030458_

Round 1

Reviewer 1 Report

Congratulations on a very nicely performed study and written paper. The aim is clear, it is easy to follow the method and statistical workings of data. The results are of interest, not only for rearing this species of sea cucumber, but for sea cucumber farming in general.

It contributes valuable scientifically tested information on how best to feed sea cucumbers, and provides data on growth rate variation for different sizes. The field of sea cucumber breeding and rearing is best developed in Asia, of which little is published in English, making this paper a valuable contribution to the field.

An added short paragraph, perhaps both in intro and discussion, regarding the effect of animal density on growth would be nice. You clearly state both number of animals and biomass per volume, so it might be interesting to  briefly discuss the results from that perspective also.

Do not forget to spell Ulva lactuca correctly on row 91 and correct the Figure legends to English.

Author Response

The authors would like to thank the reviewer for the comments, positive feedback on the study and suggestions that helped improved the manuscript for publication. Below there is a point-by-point reply to the comments.

# Reviewer 1

Congratulations on a very nicely performed study and written paper. The aim is clear, it is easy to follow the method and statistical workings of data. The results are of interest, not only for rearing this species of sea cucumber, but for sea cucumber farming in general.

It contributes valuable scientifically tested information on how best to feed sea cucumbers, and provides data on growth rate variation for different sizes. The field of sea cucumber breeding and rearing is best developed in Asia, of which little is published in English, making this paper a valuable contribution to the field.

R: Thank you for the positive feedback

An added short paragraph, perhaps both in intro and discussion, regarding the effect of animal density on growth would be nice. You clearly state both number of animals and biomass per volume, so it might be interesting to  briefly discuss the results from that perspective also.

R: The reviewer is right about the relevance of density on growth and this information had not been included in the first version of the manuscript. The number of individuals, always below 1 kg/m2, was selected to avoid density effects, so that would not be a confounding effect. So, that information and supporting reference was added to the M&M section (lines 104-105). Since density is not an effect in the trial, the authors feel that this discussion would be somewhat off-topic and throw the reader in a different direction.

Do not forget to spell Ulva lactuca correctly on row 91 and correct the Figure legends to English.

R: All corrections were made accordingly

Reviewer 2 Report

Review

Paper title: Effect of enriched substrate on the growth of the sea cucumber Holothuria arguinensis Koehler & Vaney, 1906 juveniles

The authors conducted a laboratory study to reveal the effects of enriched substrate on the growth performance of the sea cucumber Holothuria arguinensis in Portugal. The authors found positive effects when the enriched substrate was added in comparison to non-enriched substrate indicating that this method is promising for the aquaculture industry. This study may have important implications for the development of Holothuria arguinensis rearing techniques.

All these reasons explain the relevance of the paper by Tiago Rodrigues and co-authors submitted to "Diversity".

General scores.

The data presented by the authors are original and significant. The study is correctly designed and the authors used appropriate sampling methods. In general, statistical analyses are performed with good technical standards. The authors conducted careful work that may attract the attention of a wide range of specialists focused on sea cucumber aquaculture.

Recommendations.

The authors should use the MDPI style for citations and references.

The authors should mention the location of the laboratory where the study was undertaken.

Specific remarks.

L 27. Consider replacing “ability for” with “ability of”

L 62. Consider replacing “with a specific focus on using the enriched substrate as a via to provide the food” with “out with a specific focus on using the enriched substrate as a via to provide food”

L 105. Consider replacing “begin the trail” with “begin the trial”

L 106. Consider replacing “sediment in” with “sediment on”

L 113. Consider replacing “macroalgae was” with “macroalgae were”

L 126. Consider replacing “the system C” with “System C”

L 139. Consider replacing “around 90% of the water, with both parts being renew” with “as around 90% of the water, with both parts being renewed”

L 148. Consider replacing “with presence” with “with the presence”

L 150. Consider replacing “the systems A and B, and new sand in the C” with “systems A and B, and new sand in C”

L 188. Consider replacing “Although, no” with “Although no”

L 213, Fig. 2 Consider replacing “Classe” with “Class” on the two upper plates.

Fig. 2 caption. The Latin name “Holothuria arguinensis” should be italicized

L 215. Consider replacing “size-classes” with “size classes”

L 217. Consider replacing “the system” with “system”

L 218. Consider replacing “the system” with “system”

L 227. Consider replacing “the system” with “system”

Fig. 4 caption. The Latin name “Holothuria arguinensis” should be italicized

Fig. 5 Consider replacing “dia” with “day”.

L 250. Consider replacing “size-class” with “size class”

Fig. 6 caption. The Latin name “Holothuria arguinensis” should be italicized

L 258. Consider replacing “size” with “sizes”

L 265. Consider replacing “cucumbers physiology. Despite this, the mortality registered in the control system were” with “cucumber physiology. Despite this, the mortality registered in the control system was”

L 269. Consider replacing “an enriched sediment with Sargassum sp. showed a significantly” with “enriched sediment with Sargassum sp. showed significantly”

L 272. Consider replacing “survival” with “survival rate”

L 273. Consider replacing “size-class” with “size class”

L 274. Consider replacing “size-class” with “size class”

L 275. Consider replacing “60-days” with “60-day”

L 276. Consider replacing “The high survival” with “A high survival”

L 278. Consider replacing “perform by” with “performed by”

L 282. Consider replacing “class 2 and 3” with “classes 2 and 3”

L 289. Consider replacing “most of studies” with “most studies”

L 290. Consider replacing “growth” with “growth rate”

L 293. Consider replacing “where” with “whereas”

L 294. Consider replacing “previous mentioned” with “above mentioned”

L 296. Consider replacing “previous mentioned studies” with “previous studies”

L 302. Consider replacing “explanation to” with “explanation for”

L 316. Consider replacing “Saragssum  sp.” with “Sargassum  sp.”

L 322. Consider replacing “walls in” with “walls on”

L 325. Consider replacing “Relative standard deviation” with “The relative standard deviation”

L 326. Consider replacing “the system” with “system”

L 328. Consider replacing “the system” with “system”

L 329. Consider replacing “in” with “on”

L 330. Consider replacing “in” with “on”

L 331. Consider replacing “like demonstrated in” with “as demonstrated by”

L 336. Consider replacing “In this study has been demonstrated” with “This study demonstrated”

Author Response

The authors would like to thank the reviewer for the comments and suggestions for corrections that helped improved the manuscript for publication. Below there is a point-by-point reply to the comments.

# Reviewer 2

The authors conducted a laboratory study to reveal the effects of enriched substrate on the growth performance of the sea cucumber Holothuria arguinensis in Portugal. The authors found positive effects when the enriched substrate was added in comparison to non-enriched substrate indicating that this method is promising for the aquaculture industry. This study may have important implications for the development of Holothuria arguinensis rearing techniques.

All these reasons explain the relevance of the paper by Tiago Rodrigues and co-authors submitted to "Diversity".

General scores

The data presented by the authors are original and significant. The study is correctly designed and the authors used appropriate sampling methods. In general, statistical analyses are performed with good technical standards. The authors conducted careful work that may attract the attention of a wide range of specialists focused on sea cucumber aquaculture.

Recommendations

The authors should use the MDPI style for citations and references.

R: That change was made accordingly.

The authors should mention the location of the laboratory where the study was undertaken.

R: The lab name and research centre was added

Specific remarks

L 27. Consider replacing “ability for” with “ability of”

R: The correction was made as suggested

L 62. Consider replacing “with a specific focus on using the enriched substrate as a via to provide the food” with “out with a specific focus on using the enriched substrate as a via to provide food”

R: The correction was made as suggested

L 105. Consider replacing “begin the trail” with “begin the trial”

R: The correction was made as suggested

L 106. Consider replacing “sediment in” with “sediment on”

R: The correction was made as suggested

L 113. Consider replacing “macroalgae was” with “macroalgae were”

R: The correction was made as suggested

L 126. Consider replacing “the system C” with “System C”

R: The correction was made as suggested

L 139. Consider replacing “around 90% of the water, with both parts being renew” with “as around 90% of the water, with both parts being renewed”

R: The correction was made as suggested

L 148. Consider replacing “with presence” with “with the presence”

R: The correction was made as suggested

L 150. Consider replacing “the systems A and B, and new sand in the C” with “systems A and B, and new sand in C”

R: The correction was made as suggested

L 188. Consider replacing “Although, no” with “Although no”

R: The correction was made as suggested

L 213, Fig. 2 Consider replacing “Classe” with “Class” on the two upper plates.

R: The correction was made as suggested

Fig. 2 caption. The Latin name “Holothuria arguinensis” should be italicized

R: The correction was made as suggested

L 215. Consider replacing “size-classes” with “size classes”

R: The correction was made as suggested

L 217. Consider replacing “the system” with “system”

R: The correction was made as suggested

L 218. Consider replacing “the system” with “system”

R: The correction was made as suggested

L 227. Consider replacing “the system” with “system”

R: The correction was made as suggested

Fig. 4 caption. The Latin name “Holothuria arguinensis” should be italicized

R: The correction was made as suggested

Fig. 5 Consider replacing “dia” with “day”.

R: The correction was made as suggested

L 250. Consider replacing “size-class” with “size class”

R: The correction was made as suggested

Fig. 6 caption. The Latin name “Holothuria arguinensis” should be italicized

R: The correction was made as suggested

L 258. Consider replacing “size” with “sizes”

R: The correction was made as suggested

L 265. Consider replacing “cucumbers physiology. Despite this, the mortality registered in the control system were” with “cucumber physiology. Despite this, the mortality registered in the control system was”

R: The correction was made as suggested

L 269. Consider replacing “an enriched sediment with Sargassum sp. showed a significantly” with “enriched sediment with Sargassum sp. showed significantly”

R: The correction was made as suggested

L 272. Consider replacing “survival” with “survival rate”

R: The correction was made as suggested

L 273. Consider replacing “size-class” with “size class”

R: The correction was made as suggested

L 274. Consider replacing “size-class” with “size class”

R: The correction was made as suggested

L 275. Consider replacing “60-days” with “60-day”

R: The correction was made as suggested

L 276. Consider replacing “The high survival” with “A high survival”

R: The correction was made as suggested

L 278. Consider replacing “perform by” with “performed by”

R: The correction was made as suggested

L 282. Consider replacing “class 2 and 3” with “classes 2 and 3”

R: The correction was made as suggested

L 289. Consider replacing “most of studies” with “most studies”

R: The correction was made as suggested

L 290. Consider replacing “growth” with “growth rate”

R: The correction was made as suggested

L 293. Consider replacing “where” with “whereas”

R: The correction was made as suggested

L 294. Consider replacing “previous mentioned” with “above mentioned”

R: The correction was made as suggested

L 296. Consider replacing “previous mentioned studies” with “previous studies”

R: The correction was made as suggested

L 302. Consider replacing “explanation to” with “explanation for”

R: The correction was made as suggested

L 316. Consider replacing “Saragssum  sp.” with “Sargassum  sp.”

R: The correction was made as suggested

L 322. Consider replacing “walls in” with “walls on”

R: The correction was made as suggested

L 325. Consider replacing “Relative standard deviation” with “The relative standard deviation”

R: The correction was made as suggested

L 326. Consider replacing “the system” with “system”

R: The correction was made as suggested

L 328. Consider replacing “the system” with “system”

R: The correction was made as suggested

L 329. Consider replacing “in” with “on”

R: The correction was made as suggested

L 330. Consider replacing “in” with “on”

R: The correction was made as suggested

L 331. Consider replacing “like demonstrated in” with “as demonstrated by”

R: The correction was made as suggested

L 336. Consider replacing “In this study has been demonstrated” with “This study demonstrated”

R: The correction was made as suggested